

# Hyperdimensional computing in biomedical sciences: a brief review

Fabio Cumbo[1,*] and Davide Chicco[2,3,*]

[1] Center for Computational Life Sciences, Lerner Research Institute, Cleveland Clinic Foundation, Cleveland, Ohio, United States
[2] Dipartimento di Informatica Sistemistica e Comunicazione, Università di Milano-Bicocca, Milan, Italy
[3] Institute of Health Policy Management and Evaluation, University of Toronto, Toronto, Canada
* These authors contributed equally to this work.

## ABSTRACT

Hyperdimensional computing (HDC, also known as vector-symbolic architectures—VSA) is an emerging computational paradigm that relies on dealing with vectors in a high-dimensional space to represent and combine every kind of information. It finds applications in a wide array of fields including bioinformatics, natural language processing, machine learning, artificial intelligence, and many other scientific disciplines. Here we introduced the basic foundations of the HDC, focusing on its application to biomedical sciences, with a particular emphasis to bioinformatics, cheminformatics, and medical informatics, providing a critical and comprehensive review of the current HDC landscape, highlighting pros and cons of applying this computational paradigm in these specific scientific domains. In this study, we first selected around forty scientific articles on hyperdimensional computing applied to biomedical data existing in the literature, and then analyzed key aspects of their studies, such as vector construction, data encoding, programming language employed, and other features. We also counted how many of these scientific articles are open access, how many have public software code available, how many groups of authors, journals, and conferences are most present among them. Finally, we discussed the advantages and limitations of the HDC approach, outlining potential future directions and open challenges for the adoption of HDC in biomedical sciences. To the best of our knowledge, our review is the first open brief survey on this topic among the biomedical sciences, and therefore we believe it can be of interest and useful for the readership.

Corresponding authors
Fabio Cumbo, cumbof@ccf.org
Davide Chicco, davidechicco@davidechicco.it

## INTRODUCTION

The escalating energy demands of conventional computing systems recently raised concerns regarding their long-term sustainability. As computational demands continue to escalate, it also increases the environmental impact of these energy requirements (*Katal, Dahiya & Choudhury, 2022*). Consequently, research has increasingly focused on alternative computing paradigms capable of circumventing these limitations and offering more efficient, scalable, and sustainable solutions. Hyperdimensional computing (HDC) (*Kanerva, 2009*) represents one such paradigm that has gained significant attention in

recent years, tackling complex computational challenges in various scientific domains, including biomedical sciences.

Unlike traditional computing, which relies on manipulating bits in a sequential manner, HDC draws inspiration from the way the human brain is able to represent and process information. Our brain excels at handling complex, noisy, and often incomplete information, performing tasks like pattern recognition, association, and generalization with remarkable speed and efficiency. It achieves this through a massively parallel architecture, where billions of neurons interconnected *via* trillions of synapses process information collectively (*Mehonic & Kenyon, 2022*).

HDC attempts to emulate these principles by representing data as points in a high-dimensional space, typically using vectors with thousands of dimensions. These vectors, often referred to as hypervectors, serve as a fundamental building block for representing and manipulating information. The high dimensionality of these representations offers several advantages. Firstly, it allows for a holographic representation of data, where information is distributed across the entire vector, making HDC inherently robust to noise and data corruption. Secondly, it enables the use of simple, neurally plausible operations for computation, such as vector addition, multiplication, and permutation, which can be implemented efficiently in hardware (*Aygun et al., 2023*).

## Key principles and advantages of hyperdimensional computing

At its core, HDC operates on the following key principles (*Kanerva, 2009*):

(1) High-dimensional representations: information is encoded using random binary or bipolar high-dimensional vectors, usually with 10-thousand dimensions. Note that this number is not arbitrary. In high-dimensional spaces, randomly generated vectors tend to be nearly orthogonal. This property guarantees that vectors representing different information remain distinct and easily distinguishable in the same space;

(2) Randomized encoding: data is mapped to hypervectors using randomized encoding techniques, ensuring that similar data points are mapped to nearby points in the hyperspace, regardless of the nature of data (*e.g.*, images, text, biomedical signals);

(3) Holographic representation: information is distributed across the entire hypervector, providing robustness to noise and data corruption. Instead of having localized or sparse encoding where a small subset of dimensions carries most of the information, every component of the hypervector contributes to the representation. This brings a series of advantages:

   (a) Robustness: since the information is spread out, the representation is resilient to noise or corruption in any subset of the dimensions. Even if some dimensions are altered or lost, the overall information can be recovered;

   (b) Graceful degradation: partial damage leads to a proportional decrease in the fidelity of the representation rather than a catastrophic loss;

(4) Simple operations: computation is performed using simple arithmetic operations like the element-wise addition, multiplication, and permutation of vectors. These operations are at the base of the so called Multiply-Add-Permute (MAP) model, and each of them carries different and unique mathematical properties:

a. Binding $\otimes$: it is implemented as element-wise multiplication (or, for binary hypervectors, XOR). It brings the following mathematical properties:

  (i) Invertibility: the binding operation is approximately invertible. It means that, given $c = a \otimes b$, both $a$ and $b$ can be recoverable;

  (ii) Associativity and commutativity: the order of binding operations does not affect the result.

b. Aggregation/bundling $\oplus$: it aggregates multiple hypervectors to form a single hypervector representation of a set or group of items $c = a \oplus b$. It is implemented as the element-wise sum followed by a normalization to make sure that the resulting hypervector remains in the same representational space as the original hypervectors. It is characterized by the following properties:

  (i) Superposition principle: The bundling operation distributes information from all contributing hypervectors across the resulting hypervector in a way that retains key features of the original set. While individual hypervectors are not perfectly preserved, their influence remains in a distributed manner that allows to recognize and retrieve patterns from the aggregated representation;

  (ii) Noise tolerance: because the operation involves summing up many high-dimensional vectors, the signal of interest is maintained while random noise tends to cancel out due to averaging effects.

c. Permutation $p$: it is used to introduce order or structure into the hypervector representations. A permutation operation $p$ acts as a fixed, bijective reordering of the dimensions of a hypervector, shifting elements by a fixed number of positions. It also brings the following properties:

  (i) Invertibility: since a permutation is a bijective mapping, it is invertible. This means that the original order of the elements can be recovered by applying the inverse permutation, or shifting elements back to the same number of positions initially used to build the permuted representation of the hypervector;

  (ii) Preservation of similarity: although permutation changes the positions of the elements, it preserves the overall distribution and statistical properties of the hypervectors.

The set of hypervectors, together with the encoding logic and its arithmetic operations on vectors, is called vector-symbolic architecture.

These principles give rise to several advantages that make HDC particularly well-suited for a wide range of applications:

- Robustness to noise and errors: the distributed nature of information in hypervectors makes models encoded through the HDC paradigm tolerant to noise and errors;
- Computational efficiency: simple vector operations can be implemented efficiently in hardware, leading to fast and energy-efficient computation. An example is the development of an FPGA-based accelerator for HDC, which achieved significant speedups and energy savings compared to traditional CPU implementations. This makes HDC suitable for real-time applications in resource-constrained environments (*Chen, Barkam & Imani, 2023*);
- Data agnosticism: HDC can be applied to any kind of data, regardless of its nature, making it extremely versatile and suitable for a wide range of applications, ranging from robotics (*Hassan et al., 2024*), to natural language processing (*Berster, Caleb Goodwin & Cohen, 2012*), image recognition (*Neubert & Schubert, 2021*), and many others, demonstrating the HDC's ability to handle diverse data sources in very different and specific contexts efficiently;
- Scalability: the inherent parallelism of HDC allows for efficient scaling to handle large datasets and complex computations. It is crucial for applications involving big data and real-time processing (*Heddes et al., 2024*).

### The *Dollar of Mexico* and the promise of hyperdimensional computing

The foundation of HDC is rooted in the idea that information can be represented and manipulated using high-dimensional vectors, typically consisting of thousands of dimensions. One of the earliest illustrations of this concept is the "What's the Dollar of Mexico?" problem (*Kanerva, 2009*), introduced by Pentti Kanerva to demonstrate how distributed representations can encode semantic relationships. The problem states that if a system knows the relationships *Dollar → USA* and *Peso → Mexico*, it should infer that *Dollar of Mexico* likely refers to *Peso*, despite never having encountered this specific phrase before.

Mathematically, this is achieved using vector-symbolic architectures (VSAs), where concepts are represented as high-dimensional random vectors, and relationships are encoded through arithmetic operations such as vector binding and bundling as discussed above. Given the vector representations of the concepts *Currency* and *Country*, and *Dollar* and *USA*, their relationship can be represented as:

$$V_{USA} = (Currency \otimes Dollar) \oplus (Country \otimes USA),$$

where $\otimes$ and $\oplus$ denote the binding and bundling operations, respectively. Analogously, the relationship between *Peso* and *Mexico* can be defined as $V_{Mexico} = (Currency \otimes Peso) \oplus (Country \otimes Mexico)$. Finally, the binding between $V_{USA}$ and $V_{Mexico}$ produces a new vector that represents the reality that we are trying to encode. In order to answer the initial question, a simple binding between this final vector with the

vector representation of *Dollar* produces a new vector which is very similar to the vector representation of *Peso*. This can be verified by computing the cosine similarity between this final vector and all the other vectors computed so far.

This principle—where information is stored and retrieved based on patterns of similarity rather than explicit lookup tables—forms the backbone of HDC. Over the years, researchers have leveraged these properties to build robust models capable of handling noisy, uncertain, and large-scale data, making HDC particularly appealing for different kinds of problems, including biomedical applications.

It represents a paradigm shift in computing, offering a powerful and efficient alternative to traditional approaches. Its unique ability to handle high-dimensional data, robustness to noise, and computational efficiency make it a promising candidate for addressing the challenges posed by the ever-growing complexity and scale of data-driven applications. As research in HDC continues to advance, we can expect to see its adoption in an even wider range of fields, paving the way for a new era of intelligent and efficient computing systems.

## Hyperdimensional computing and biomedical sciences

While a broader potential of HDC is clear, its application to biomedical sciences presents a particularly compelling case. This review focuses specifically on the intersection of HDC and biomedical sciences, encompassing medical informatics, bioinformatics, and cheminformatics. As illustrated in Fig. 1, the number of publications exploring HDC within these domains has steadily increased over the past decade (with the exception of 2019 and 2023), highlighting the growing interest and potential of this technology in these specific scientific domains.

This review is particularly crucial due to the rapid evolution and relative novelty of HDC in the biomedical domain. While promising, this field is still in its early stage, characterized by a diversity of approaches and a lack of standardized methodologies.

The motivations for this review systems from several key factors that distinguish it from previous surveys, particularly those with a general or solely technical focus:

(1) The biomedical applications of HDC are evolving at an unprecedented rate, with novel approaches and implementations emerging frequently. While previous surveys have outlined the theoretical and algorithmic foundations of HDC (*Stock et al., 2024*), this review captures the latest trends and breakthroughs specific to biomedical sciences. It offers an updated perspective that reflects the current research landscape, making it indispensable for readers who wish to stay abreast of state-of-the-art developments;

(2) As mentioned above, unlike prior surveys that address HDC from a broad, interdisciplinary viewpoint, this paper zooms in on biomedical applications. By bridging the gap between general HDC methodologies and domain-specific needs, this review serves as a specialized resource for biomedical researchers and practitioners;

(3) The field of HDC in biomedicine is still in its nascent stages, characterized by a diversity of approaches and a lack of standardized methodologies. This review synthesizes the existing literature to help readers identifying promising avenues for future innovations in this field;

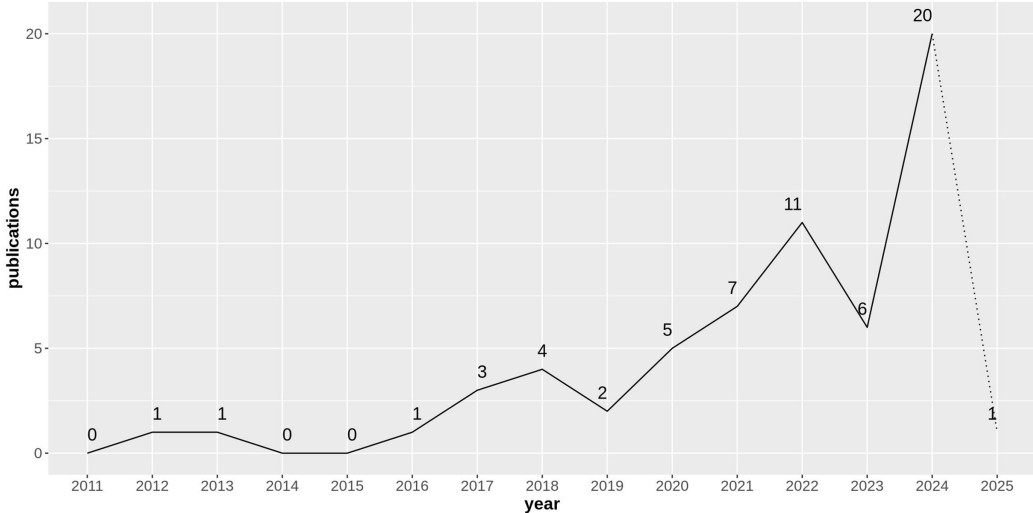

**Figure 1 Number of research articles considered in this survey on hyperdimensional computing applied to biomedical sciences published over time.** Note that 2025 refers only to January and the first half of February.

Our target audience encompasses both researchers actively working in HDC and those in related biomedical fields seeking to explore the potential of this novel computational paradigm. By providing them with a critical evaluation of the current landscape, as well as identifying gaps and future directions, this review aims to equip readers with the knowledge and insights needed to advance HDC applications in biomedical sciences.

In summary, this article offers a unique contribution by focusing specifically on the intersection of HDC and biomedicine, delivering a synthesis of recent progress, a critical evaluation of existing challenges, and practical guidance for future research. For readers seeking to understand not only the theoretical foundations but also the practical potential and future trajectory of HDC in biomedical sciences, this review serves as an essential resource that complements and extends previous surveys.

To the best of our knowledge, no surveys on biomedical applications of HDC exist in the scientific literature. The already-mentioned study by *Stock et al. (2024)* recently presented a technical guide or introduction aimed at explaining the fundamentals of HDC and demonstrating its potential through specific applications in bioinformatics. In particular, their article is structured to provide an introduction to the HDC concepts, outline the computational advantages of HDC from a general perspective, and highlight selected case studies and applications across the bioinformatics field. Although useful and interesting, this article does not outline the applications of HDC in biomedical sciences; we fill this gap by presenting our short survey here.

## SURVEY METHODOLOGY

Our literature review aims to provide a comprehensive overview of the current state of hyperdimensional computing in biomedical sciences. To ensure a systematic and rigorous

approach, we adopted guidelines outlined in *Pautasso*'s *(2013)* rules on writing a literature review throughout the survey methodology.

## Search strategy and selection criteria

Our bibliographic research employed a systematic search strategy across Google Scholar and the collection of scientific articles in the *hd-computing* repository at https://www.hd-computing.com. The latter represents a valuable resource to help the HDC/VSA scientific community by collecting software, video courses and webinars, in addition to scientific articles about the application of the HDC paradigm to specific technological problems.

To ensure a comprehensive and unbiased review, we conducted a systematic search of the literature using a multi-pronged approach. In particular, we utilized a combination of keywords and their variations to ensure a broad capture of relevant articles among those operating in the context of biomedical sciences: *("Hyperdimensional Computing" OR "Vector-Symbolic Architectures") AND ("Bioinformatics" OR "Cheminformatics" OR "Medical Informatics" OR "Biomedical Sciences")*

Our search yielded a pool of 41 articles. To focus our review on the most relevant studies, we applied the following inclusion criteria:

- Focus on HDC: articles must primarily focus on the application of HDC techniques;
- Relevance to biomedical sciences: articles must address problems within bioinformatics, cheminformatics, or medical informatics;
- Peer-reviewed publications: we prioritized peer-reviewed journal articles and conference proceedings articles. However, we also considered preprints of significant interest and potential impact.

## Data extraction and categorization

We extracted a comprehensive set of features from each selected article in support of our analysis of HDC applications in biomedical sciences. These features were carefully chosen to provide insights into various aspects of the research, including:

(1) Bibliographic information:

- Theme: categorized the primary application domain of the article into bioinformatics, cheminformatics, or medical informatics;
- Venue: recorded the journal or conference name where the article was published;
- Article type: classified the publication type as a journal article, conference proceedings article, or preprint;
- Open access: noted whether the article was freely available in an open-access format.

(2) HDC methodology:

- Vector construction: documented the specific methods used to construct the hypervectors, such as random projection, learned embeddings, or domain-specific encodings;

- Data encoding: described the techniques employed to encode different data types (*e.g.*, DNA sequences, molecular structures, patient records) into hypervectors;
- Data combination: analyzed the operations used to combine encoded data, including vector addition, multiplication, and permutation operations.

(3) Application and evaluation:

- Scientific problem: summarized the specific biomedical problem addressed in the article;
- Number of patients and samples: recorded the dataset size, including the number of patients or samples used in the study;
- Hardware: extracted information about any specialized hardware utilized for HDC implementation, such as classical CPUs, GPUs, ASICs, or FPGAs;
- Robustness: analyzed any reported metrics or discussions related to the model's robustness to noise;
- Programming language(s): identified the programming language(s) used for implementing HDC models (for example, Python, MATLAB, R, C/C++, and Rust);
- Software availability: determined whether the software or code developed for the study was publicly available.

## Analysis framework

The extracted data allowed us to analyze the trends and characteristics of HDC applications in biomedical sciences. We will present our findings by focusing on each of the extracted features, highlighting the following aspects:

- Themes: we will discuss the prevalence of HDC in each of the three themes (bioinformatics, cheminformatics, and medical informatics), identifying areas where HDC has been particularly successful or is gaining traction;
- Methodological Trends: we will analyze the commonalities and differences in HDC methodologies employed across different studies, examining the suitability of specific techniques for particular biomedical problems;
- Open Science Practices: we will investigate the adoption of open science practices, such as open access publication and software sharing, within the HDC community.

By systematically analyzing these features, this literature review aims to provide a comprehensive and insightful overview of the current landscape of HDC in biomedical sciences, identifying research gaps, and highlighting future directions for this promising field.

We identified a pool of 62 research articles over three scientific themes, *i.e.*, medical informatics (*Buteau et al., 2024*; *Chen et al., 2024a*, *2024b*; *Du et al., 2024*; *Gaddi et al., 2024*; *Ponzina et al., 2024*; *Salerno & Barraud, 2024*; *Cohen et al., 2012*; *Moon et al., 2013*; *Kleyko et al., 2016*, *2017*; *Rahimi et al., 2017a*, *2017b*, *2018*; *Burrello et al., 2018*, *2019a*, *2019b*, *2021*; *Lagunes & Lee, 2018*; *Burkhardt et al., 2019*; *Asgarinejad, Thomas & Rosing,*

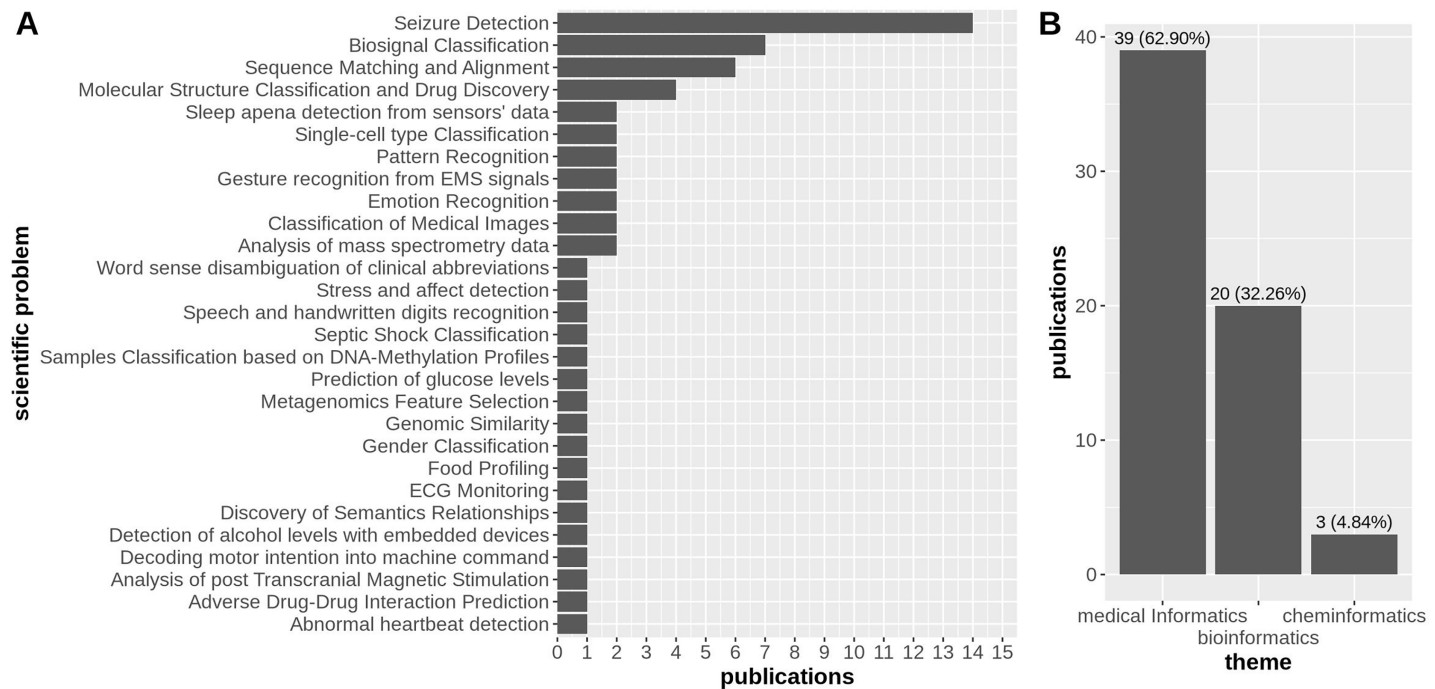

**Figure 2** Distribution of publications over 15 scientific problems discussed in the research articles considered in this survey (A), and the distribution of research articles across the three areas of biomedical sciences, *i.e.*, medical informatics, bioinformatics, and *cheminformatics*, with a strong prevalence of articles falling into the first category (B).

*2020*; *Billmeyer & Parhi, 2021*; *Ge & Parhi, 2021*, *2022*; *Menon et al., 2021*, *2022*; *Pale, Teijeiro & Atienza, 2021*, *2022a*, *2022b*, *2022c*, *2023*; *Watkinson et al., 2021a*, *2021b*; *Schindler & Rahimi, 2021*; *Ni et al., 2022*; *Shahroodi et al., 2022*; *Ung, Ge & Parhi, 2022*; *Wang, Ma & Jiao, 2022*; *Topic et al., 2022*; *Segura et al., 2024*; *Ge et al., 2024*; *Xu & Parhi, 2024*; *Katoozian, Hosseini-Nejad & Dehaqani, 2024*; *Jeong et al., 2024*; *Colonnese et al., 2025*), bioinformatics (*Barmpas et al., 2024*; *Fan et al., 2024*; *Mohammadi et al., 2024*; *Pinge et al., 2024*; *Imani et al., 2018*; *Kim et al., 2020*; *Cumbo, Cappelli & Weitschek, 2020*; *Poduval et al., 2021*; *Chen & Imani, 2022*; *Zou et al., 2022*; *Barkam et al., 2023*; *Verges et al., 2024*; *Xu et al., 2024*; *Cumbo et al., 2024*), and cheminformatics (*Ma, Thapa & Jiao, 2022*; *Jones et al., 2023*, *2024*), with 32 of them falling under the medical informatics category (78.05%), 7 belonging to the bioinformatics domain (17.07%), and the remaining 2 about cheminformatics (4.88%) (Fig. 2B).

In the context of three macro thematic categories, we manually classified the whole set of 62 articles based on the specific scientific problem that the authors aimed at addressing with their research. In particular, we identified 28 scientific problems, with the most discussed being seizure detection (14/39 articles under the medical informatics theme), followed by biosignal classification and sequence matching and alignment with roughly the same number of manuscripts (7/39 articles under the medical informatics theme, and 6/20 articles under the bioinformatics theme, respectively) (Fig. 2A).

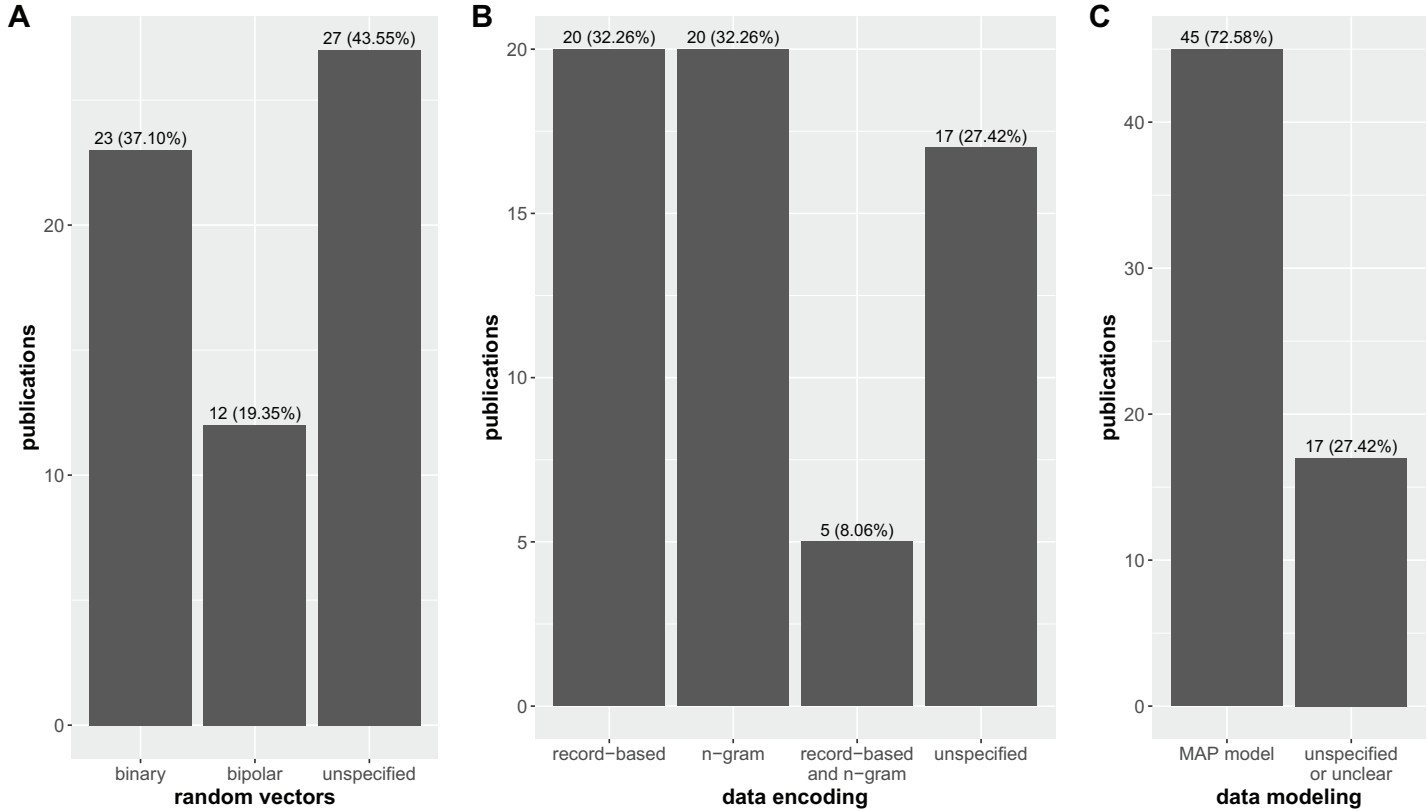

**Figure 3 Distribution of research articles based on the type of random vectors (*i.e.*, binary, bipolar, and unspecified–A), data encoding technique (*i.e.*, record-based, n-gram, record-based and n-gram, and unspecified–B), and data modeling method (MAP model and unspecified or unclear–C) used for building the HDC architecture over the set of manuscripts considered in this study.**

Examining the methodological choices employed across the surveyed literature reveals a diverse landscape with a few dominant trends and some concerning ambiguities (Fig. 3).

In particular, Fig. 3A highlights the prevalence of random vector generation methods, with ~50% of the studies utilizing either binary (23 articles) or bipolar (12 articles), and the other ~50% failing to explicitly specify their approach (27 articles), making it difficult to assess the potential impact of vector generation choices on the reported results.

Analogously, Fig. 3B shows the different data encoding techniques employed in all the 62 studies, revealing a perfect balance between record-based encoding (20 articles) and n-gram representations (20 articles). Interestingly, a small subset (5 articles) employs a combination of both. However, a concerning number of studies (17 articles) again lack specificity in describing their encoding strategy.

Finally, Fig. 3C examines data modeling techniques, revealing a striking dominance of the MAP-model (45 articles). This finding suggests a potential convergence towards this approach within the HDC community, at least in the context of biomedical applications. However, a substantial number of studies (17 articles) also fails in clearly specifying the employed modeling technique.

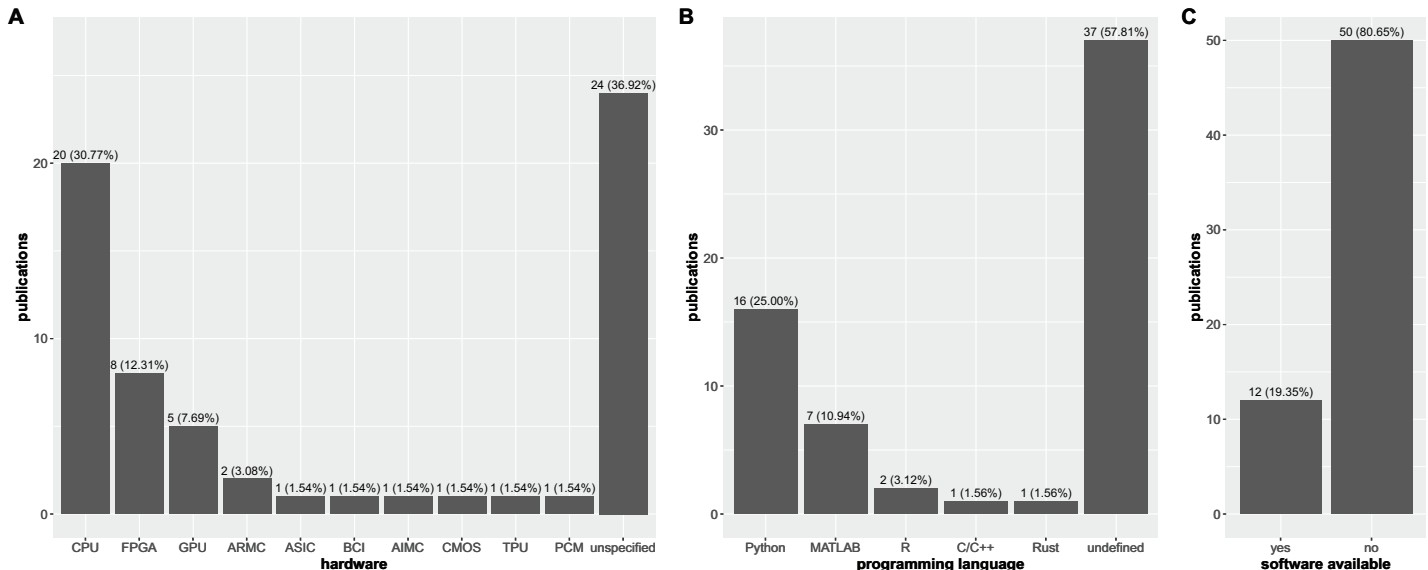

**Figure 4 Distribution of software implementations over the different types of hardware architectures, programming languages, and software availability.** Distribution of software implementations described in the research articles considered in this article over the different types of hardware architectures (*i.e.*, *CPU, FPGA, GPU, ARMC, ASIC, BCI, AIMC, CMOS, TPU, PCM*, and *unspecified*–A) for which they have been specifically designed (note that the same implementation could be available for more than a single type of hardware) together with the distribution of the same number of research articles over the programming languages used to develop the HDC models (B) and the number of articles for which authors made their software available (C).

It is worth noting that one critical aspect that still remains largely unexplored in the literature considered in this review is the role of randomness in hypervector generation and its potential impact on both model accuracy and hardware efficiency. This lack of investigation is likely due to the intrinsic nature of HDC, where information is encoded into randomly generated hypervectors by design. Since randomness is a fundamental principle of HDC, it is very likely that authors implicitly accept it without examining how different randomness strategies might affect accuracy or computational efficiency. However, vector generation and encoding are among the most computationally expensive steps in an HDC system, and understanding potential trade-offs between randomness, accuracy, and hardware performance could be particularly valuable specifically for biomedical applications, where precision and resource constraints are critical. Future research should explore whether alternative encoding methods—such as semi-random or structured approaches—could improve performance while maintaining the robustness of HDC models.

It is also worth to note that ~40% of the research articles considered in this study that propose a HDC-based software solution also do not mention the hardware architecture for which their software has been designed to work on (Fig. 4A). Furthermore, the limited exploration of specialized hardware like ASICs and neuromorphic systems indicates a potential gap in leveraging the full potential of HDC for computationally demanding biomedical tasks.

As a direct consequence, when the authors do not mention the specific hardware for which their software has been designed and tested, they also usually omit the programming language used to develop their software (Fig. 4B). For all the other cases, Python is the most used programming language in this context, followed by MATLAB, and finally R, C/C++, and Rust with no more than two references each.

A lack of transparency and reproducibility is evident in the alarming trend of withholding the source code used to design and build the HDC models in these specific scientific fields of biomedical sciences (Fig. 4C). This lack of openness represents a fundamental breach of scientific principles, severely hindering the progress and credibility of the field and raising concerns about the thoroughness and scientific rigor of these studies, clearly standing in the opposite direction with respect to the core values of FAIR principles (*Wilkinson et al., 2016*).

More importantly, the refusal to share source code is even more concerning considering that, in most of the cases, authors have taken the time to assign specific names to their software, highlighting a deliberate choice towards obfuscation rather than openness. Without access to the code, the scientific community is left to accept results on blind faith. Independent verification becomes impossible, hindering the identification of potential errors, biases, or overstated claims. This practice stifles collaborative progress, as researchers are unable to build upon existing work, forcing them to reinvent the wheel or resort to potentially less efficient solutions.

Analyzing the publication venues of HDC research in biomedical sciences reveals a significant skew towards conference proceedings over peer-reviewed journals (Table 1 for a list of journals and number of published articles, and Table 2 for a list of conferences alongside the number of articles published in their proceedings and their publisher). While it is generally common in rapidly evolving fields, this trend raises concerns one more time about the long-term impact and acceptance of HDC within the broader scientific community.

As it is evident from this data, a striking majority of publications appear in conference proceedings, with IEEE conferences dominating the landscape. While conferences offer a valuable platform for a rapid dissemination of research findings, they often lack the rigorous peer-review processes typical of academic journals.

The disparity in publication venues raises concerns about the rigor and transparency of HDC research in biomedical sciences. The fact that the number of articles published in conference proceedings (34) significantly outweigh those published in peer-reviewed journals (21) suggests a potential lack of peer-review scrutiny (Fig. 5A).

In fact, conference proceedings publications allow for a faster spread of new ideas, but they have several drawbacks compared to journal publications. Typically, the peer review process for conference proceedings is more superficial, as conference organizers need to accept a certain number of articles to cover the costs of organizing the conference. Additionally, conference proceedings are usually short (4–8 pages), which limits the space available for demonstrations, specific experiments, and in-depth scientific discussions (*Ernst, 2006*).

**Table 1 Ranking of journals and preprint servers per number of published articles.**

| Journal | Publisher | Articles |
| --- | --- | --- |
| arXiv preprint | Cornell University | 6 |
| Scientific Reports | Springer Nature | 2 |
| Journal of Signal Processing Systems | Springer Nature | 2 |
| IEEE Transactions on Biomedical Engineering | IEEE | 2 |
| Frontiers in Neurology | Frontiers | 2 |
| Algorithms | MDPI | 1 |
| Bioinformatics | Oxford University Press | 1 |
| biorXiv preprint | Cold Spring Harbor Laboratory | 1 |
| IEEE Access | IEEE | 1 |
| IEEE Design & Test | IEEE | 1 |
| IEEE Embedded Systems Letters | IEEE | 1 |
| IEEE Internet of Things Journal | IEEE | 1 |
| IEEE Journal of Biomedical and Health Informatics | IEEE | 1 |
| IEEE Journal on Exploratory Solid-State Computational Devices and Circuits | IEEE | 1 |
| IEEE Open Journal of Circuits and Systems | IEEE | 1 |
| IEEE Transactions on Biomedical Circuits and Systems | IEEE | 1 |
| IEEE Transactions on Neural Networks and Learning Systems | IEEE | 1 |
| Journal of Biomedical Informatics | Elsevier | 1 |
| Mobile Networks and Applications | Springer Nature | 1 |
| Proceedings of the IEEE | IEEE | 1 |
| Sensors | MDPI | 1 |

**Table 2 Ranking of conferences per articles published in their proceedings.**

| Conference | Publisher | Articles |
| --- | --- | --- |
| Annual International Conference of the IEEE Engineering in Medicine & Biology Society (EMBC) | IEEE | 7 |
| Design, Automation and Test in Europe Conference and Exhibition (DATE) | IEEE | 4 |
| IEEE Biomedical Circuits and Systems Conference (BioCAS) | IEEE | 3 |
| AMIA Annual Symposium Proceedings | AMIA | 2 |
| Asilomar Conference on Signals, Systems, and Computers | IEEE | 2 |
| IEEE International Conference on Bioinformatics and Bioengineering (BIBE) | IEEE | 2 |
| IEEE International Conference on Bioinformatics and Biomedicine (BIBM) | IEEE | 2 |
| Proceedings on the IEEE/ACM International Conference on Computer-Aided Design | IEEE | 2 |
| ACM/IEEE Design Automation Conference (DAC) | IEEE | 1 |
| IEEE Annual International Symposium on Field-Programmable Custom Computing Machines (FCCM) | IEEE | 1 |
| IEEE Congress on Evolutionary Computation (CEC) | IEEE | 1 |
| IEEE EMBS International Conference on Biomedical and Health Informatics (BHI) | IEEE | 1 |

(Continued)

| Table 2 (continued) | | |
| --- | --- | --- |
| **Conference** | **Publisher** | **Articles** |
| IEEE International Conference on Acoustics, Speech and Signal Processing (ICASSP) | IEEE | 1 |
| IEEE International Symposium on Biomedical Imaging (ISBI) | IEEE | 1 |
| Proceedings of the Annual International Symposium on Computer Architecture | ACM | 1 |
| Proceedings of the EAI International Conference on Bio-inspired Information and Communications Technologies (formerly BIONETICS) | ACM | 1 |

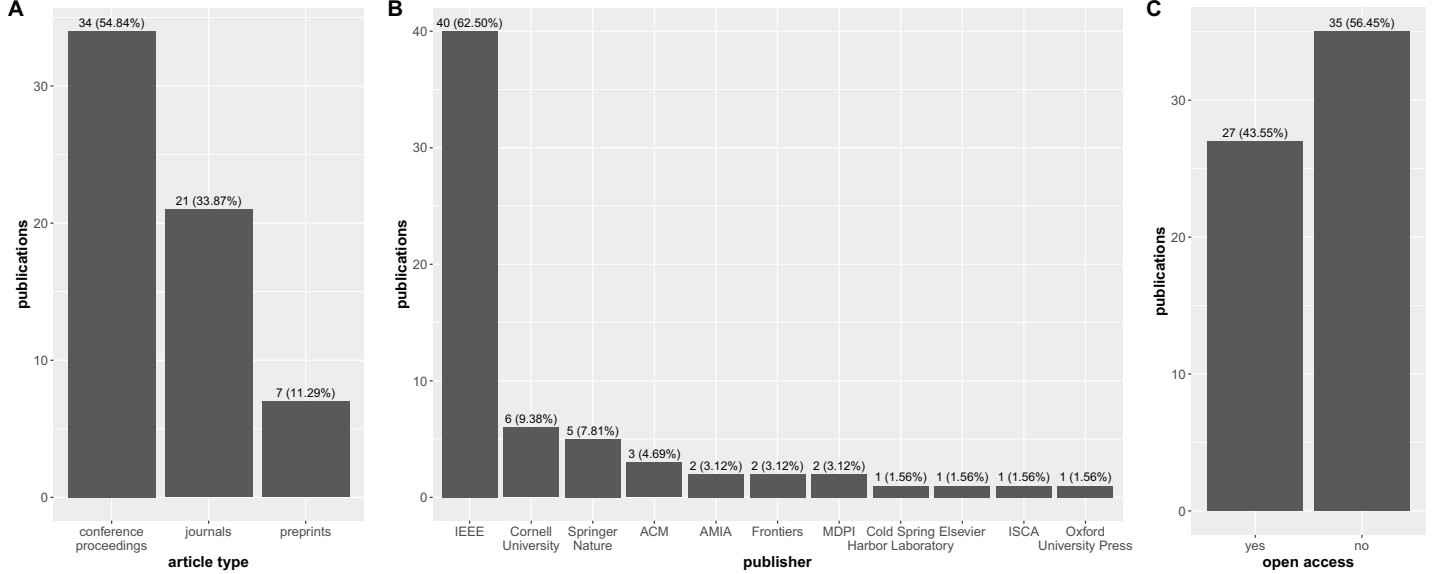

**Figure 5 Bar graph with the distribution of the 62 research articles considered in this study over the publication venues.** (A) Conference proceedings, journals, and preprints; (B) publishers; (C) open access.

Moreover, journal articles have a stronger impact on scientific literature than conference proceedings (*Lisée, Larivière & Archambault, 2008*). As Cynthia Lisée and coauthors explain: "*The evidence thus shows that conference proceedings have a relatively limited scientific impact, on average representing only about 2% of total citations, that their relative importance is shrinking, and that they become obsolete faster than the scientific literature in general*" (*Lisée, Larivière & Archambault, 2008*).

Additionally, Fig. 5B highlights the overwhelming influence of IEEE (40 articles) as the primary publisher in the field. This dominance, while indicative of the strong presence of HDC research in IEEE conferences, also raises concerns about potential publication biases and the need for greater diversity in dissemination channels.

Most concerning, however, is the stark disparity in open access publication revealed in Fig. 5C. A mere 27 articles are classified as open access, compared to a staggering 35 articles that are not freely available.

**Table 3 Ranking of authors by number of publications, with a minimum of three articles, alongside the most prevalent scientific problem discussed in their articles.** When None, specific scientific problems are discussed in one article only.

| Author | Articles | Most prevalent scientific problem |
| --- | --- | --- |
| Tajana Rosing | 10 | Sequence Matching and Alignment (2/10) |
| Abbas Rahimi | 9 | Seizure Detection (6/9) |
| Mohsen Imani | 8 | Sequence Matching and Alignment (6/8) |
| Niema Moshiri | 5 | None |
| Kaspar Schindler | 5 | Seizure Detection (5/5) |
| Luca Benini | 5 | Seizure Detection (4/5) |
| Jan Rabaey | 5 | Biosignal Classification (3/5) |
| Una Pale | 5 | Seizure Detection (4/5) |
| Tomas Teijeiro | 5 | Seizure Detection (4/5) |
| David Atienza | 5 | Seizure Detection (4/5) |
| Keshab Parhi | 5 | Biosignal Classification (2/5), Seizure Detection (2/5) |
| Alessio Burrello | 4 | Seizure Detection (4/4) |
| Lulu Ge | 4 | Seizure Detection (2/4) |
| Weihong Xu | 4 | None |
| Tony Givargis | 4 | None |
| Alexandru Nicolau | 4 | None |
| Zhuowen Zou | 3 | Sequence Matching and Alignment (2/3) |
| Trevor Cohen | 3 | None |
| Pentti Kanerva | 3 | Biosignal Classification (2/3) |
| Yeseong Kim | 3 | Sequence Matching and Alignment (2/3) |
| Jaeyoung Kang | 3 | None |
| Flavio Ponzina | 3 | None |

**Table 4 Ranking of groups of authors by number of publications.**

| Group of authors | Articles |
| --- | --- |
| David Atienza, Una Pale, Tomas Teijeiro | 5 |
| Luca Benini, Alessio Burrello, Abbas Rahimi, Kaspar Schindler | 2 |
| Lulu Ge, Keshab Parhi | 2 |
| Tony Givargis, Victor Jow, Alexandru Nicolau, Alexander Veidenbaum, Neftali Watkinson | 2 |

In our literature review, we also collected data regarding the most present single authors (Table 3) and the most present groups of authors (Table 4). As one can notice, Tajana Rosing (currently at University of California San Diego), Abbas Rahimi (currently at IBM Research Zurich), and Mohsen Imani (currently at University of California Irvine) resulted being the researchers with the highest number of authored studies in our survey, with nine and seven publications respectively (Table 3).

It is also worth noting, as we highlighted in Table 3, that the most present single authors are more leaning towards the study of three scientific problems, that are biosignal classification, seizure detection, and sequence matching and alignment.

Regarding author groups, the team consisting of David Atienza, Una Pale, and Tomas Teijeiro has resulted being quite productive, by publishing five scientific articles on HDC applied to biomedical data (Table 4). The following most present groups of authors are (i) Luca Benini, Alessio Burrello, Abbas Rahimi, and Kaspar Schindler, (ii) Lulu Ge and Keshab Parhi, (iii) and Tony Givargis, Victor Jow, Alexandru Nicolau, Alexander Vaidenbaum, and Neftali Watkinson with two published scientific articles for each group. Other groups of authors have less publications. Interestingly, Abbas Rahimi and Mohsen Imani are the most productive authors in our survey, but they do not belong to the groups of most prolific authors.

## DISCUSSION

As we observed earlier, it is clear that most applications of HDC pertain to medical informatics, with 63% of the articles related to this scientific subject (Fig. 2B). Among the 62 studies analyzed, the most frequent medical application was seizure detection, with 14 studies focusing on HDC for this purpose. Biosignal classification of EEG data and DNA sequence matching alignment are also common biomedical applications of HDC, with seven and six articles, respectively (Fig. 2A).

Regarding hardware, a large fraction of the articles (37%) did not report information, while around 31% claimed the use of HDC with CPUs (Fig. 4A).

Information about the programming languages employed in a scientific study is pivotal for its reproducibility, but most of the articles considered in our survey (37 out of 62) did not report this component (Fig. 4B). Among the articles that did include this information, Python was the most commonly used programming language. This result is not surprising, as Python is the most popular coding language worldwide according to PYPL (*PYPL Index, 2025*). Unfortunately, 81% of the studied articles did not provide their software code in public repositories, such as GitHub packages, making it impossible to reproduce their experiments. This lack of reproducibility is a major drawback of the articles considered in this survey.

Moreover, open science best practices (*Chicco, Oneto & Tavazzi, 2022*) were not followed by most authors of the 62 articles studied here; only 27 of these articles were published in open access journals, conference proceedings, or preprint servers (Fig. 5C). This is unfortunate, as most of the articles presented here cannot be read for free by individuals outside academic institutions.

No particular journal stood out as the main publishing venue for HDC in biomedicine; in fact, the arXiv preprint server was the most common publication site for long articles, with six preprints released there (Table 1). While preprint servers can be useful for disseminating new ideas, we know that their articles are not peer-reviewed, so we recommend that readers approach them with caution (*Lin et al., 2020*; *Flanagin, Fontanarosa & Bauchner, 2020*; *Zeraatkar et al., 2022*). Among conference proceedings, three conferences were the most attended by researchers applying HDC to biomedical

sciences: EMBC, DATE, and BioCAS (Table 2). Regarding publishers, IEEE produced the majority of the articles presented in this survey, publishing 63% of them (Fig. 5B).

To summarize, the review of these 62 articles clearly indicates that HDC can provide advantages for analyzing biomedical data, especially on datasets of huge dimensions. However, we have to report a general lack of information among these articles, that is clearly a drawback of the current state of HDC: as we mentioned earlier, most of the articles are not open access, do not contain links to software code repositories, and do not even state which programming languages were employed for the computational analyses. This lack of transparency makes it almost impossible to reproduce the results obtained by most of these studies.

Considering the current situation where HDC packages are developed by researchers independently, we advocate for a standardized software platform for this computational field, similar to scikit-learn for machine learning in Python (*Kramer, 2016*), Bioconductor for bioinformatics in R (*Gentleman et al., 2004*), Lux.jl for scientific machine learning in Julia (*Lux.jl, 2025*), or Linfa for machine learning in Rust (*rustdoc, 2025*).

## THE FUTURE OF HYPERDIMENSIONAL COMPUTING IN BIOMEDICAL SCIENCES

Biomedical data, ranging from genomic sequences to imaging and electronic health records, is inherently complex and heterogeneous. Traditional methods often struggle to integrate and interpret such varied information effectively (*Ziegler & Dittrich, 2007*). HDC's data-agnostic encoding, which can seamlessly transform diverse data types into a unified high-dimensional space, offers a compelling solution (*Wilkinson et al., 2016*). We strongly believe that as biomedical research increasingly focuses on multi-modal data integration, HDC will play a crucial role in developing more holistic models that can capture the intricate interplay between different biological systems (*Mehonic & Kenyon, 2022*).

Additionally, once of the major challenges in biomedical applications is the prevalence of noise and artifacts in data. The holographic and distributed nature of information encoding in HDC provides an inherent robustness that can help mitigate the effects of such noise. Moreover, the simplicity of the arithmetic operations in HDC that we presented in the introductory section above, could lead to models that are more interpretable compared for example to deep neural networks that rely on complex, multi-layered transformations that are often difficult to dissect. On the other hand, HDC models operate on high-dimensional representations through well-defined arithmetic operations such as bundling, binding, and permutation, that maintain semantic relationships between encoded features, making it possible to trace how different inputs contribute to the final decision.

We see the rising adoption of this computing paradigm in biomedical sciences as an opportunity for creating transparent, reliable computational tools that clinicians and researchers can trust, especially in critical applications like diagnostics and personalized medicine.

The scalability of HDC is another aspect that excites us. Biomedical research is generating data at an unprecedented rate, and the ability to process large-scale datasets efficiently is paramount. HDC's inherent parallelism and simple operations make it a natural fit for real-time applications, like the continuous monitoring of patient health *via* wearable devices and *ad-hoc* hardware. We strongly believe that, in the near future, advances in hardware implementations of HDC will lead to energy-efficient systems capable of real-time decision-making in clinical environments (*Zou et al., 2021*).

However, despite these promising advantages, several challenges must be addressed before HDC can be fully realized as a mainstream tool in biomedical sciences. One key limitation is the need for standardized methodologies for encoding and decoding biological data. Unlike more mature computational paradigms such as deep learning, which have well-established architectures and best practices, HDC lacks universally accepted guidelines for optimal data representation and processing pipelines. This can lead to inconsistencies in model performance across different biomedical applications (*Vergés et al., 2025*).

Furthermore, while HDC excels in handling high-dimensional representations efficiently, it may struggle with tasks that require complex hierarchical reasoning or explicit feature extraction. Many biomedical applications, such as protein structure prediction or modeling gene regulatory networks, rely on intricate spatial and temporal relationships that may not be easily captured using current HDC approaches. For instance, the survey by *Kleyko et al. (2023)* suggests that the current focus has been on simpler classification tasks, indicating a gap in addressing more intricate biomedical challenges. Expanding the expressiveness of HDC models to better accommodate these complexities remains an open challenge.

Another potential drawback is the relative immaturity of HDC in terms of software and hardware ecosystem support. Unlike deep learning, which benefits from extensive libraries and frameworks optimized for large-scale computations (*e.g.*, TensorFlow, PyTorch), HDC is still in its early stages of widespread adoption. The lack of dedicated general-purpose frameworks for vector-symbolic computing makes it difficult for researchers and developers to experiment with and deploy HDC-based solutions efficiently. However, recent efforts have emerged to bridge this gap by developing robust frameworks that simplify the implementation of HDC models, making them more accessible to a broader range of scientists and engineers (*Kang et al., 2022*; *Simon et al., 2022*; *Schlegel, Neubert & Protzel, 2021*; *Heddes et al., 2023*; *Cumbo, Weitschek & Blankenberg, 2023*).

In conclusion, we are convinced that HDC holds significant promise for revolutionizing biomedical sciences. Its ability to manage complexity, maintain robustness, and scale efficiently aligns well with the current and future demands of biomedical data analysis. However, addressing challenges related to standardization, expressiveness, and ecosystem support will be crucial for its widespread adoption. As the field progresses, we anticipate that continued innovations in both theory and practical implementations of HDC will pave the way for transformative advances in research and clinical practice, leading to a new era of intelligent, efficient, and transparent biomedical computation.

## CONCLUSIONS

Our review of HDC applications in biomedical sciences reveals a concerning trend that warrants immediate attention from the research community. While the field demonstrates significant promise, its progress is hampered by a lack of adherence to fundamental principles of scientific transparency and reproducibility.

Firstly, a significant portion of the research is published in conference proceedings rather than peer-reviewed journals. While conferences provide a valuable platform for disseminating early findings, this over-reliance on conference publications raises concerns about the rigor and validation of the research. Conference proceedings often lack the stringent peer-review processes characteristic of academic journals, potentially allowing methodological shortcomings or unsubstantiated claims to go unchallenged. This tendency towards conference publications may hinder the field's long-term growth and acceptance within the broader scientific community.

Secondly, and most alarmingly, our analysis reveals a pervasive lack of code sharing among HDC researchers in this domain. Despite the growing movement towards open science and the widely recognized FAIR principles (*Wilkinson et al., 2016*) for research outputs, the vast majority of studies fail to provide public access to the code used in their analyses. This lack of transparency is deeply troubling. Without access to the underlying code, it becomes impossible to independently verify results, replicate experiments, or build upon existing work. This practice represents a fundamental breach of scientific integrity, undermining the credibility of individual studies and hindering the collective advancement of the field.

To foster a robust and trustworthy research landscape for HDC in biomedical sciences, the adoption of open science practices is paramount. The community must prioritize publication in peer-reviewed journals and embrace a culture of code sharing. Funding agencies and journals should incentivize these practices, while researchers must recognize their ethical obligation to ensure transparency and reproducibility in their work. Failure to address these critical issues will continue to plague the field, limiting its impact and hindering the realization of HDC's full potential in revolutionizing biomedical research.

## LIST OF ABBREVIATIONS

| | |
|---|---|
| **ACM** | Association for Computing Machinery |
| **AMIA** | American Medical Informatics Association |
| **BIBE** | IEEE International Conference on Bioinformatics and Bioengineering |
| **BIBM** | IEEE International Conference on Bioinformatics and Biomedicine |
| **BioCAS** | IEEE Biomedical Circuits and Systems Conference |
| **BIONETICS** | International Conference on Bio-inspired Information and Communication Technologies |
| **BHI** | IEEE EMBS International Conference on Biomedical and Health Informatics |
| **CEC** | IEEE Congress on Evolutionary Computation |
| **CPU** | Central processing unit |

| | |
|---|---|
| **DAC** | ACM/IEEE Design Automation Conference |
| **DATE** | Design, Automation and Test in Europe |
| **EMBC** | Engineering in Medicine & Biology Society |
| **FAIR** | Findability, Accessibility, Interoperability, and Reusability |
| **FCCM** | IEEE Annual International Symposium on Field-Programmable Custom Computing Machines. |
| **GPU** | Graphics processing unit |
| **HDC** | Hyperdimensional Computing |
| **ICASSP** | IEEE International Conference on Acoustics, Speech and Signal Processing |
| **IEEE** | Institute of Electrical and Electronics Engineering |
| **ISBI** | IEEE International Symposium on Biomedical Imaging |
| **MAP** | Multiply-Add-Permute |
| **MDPI** | Multidisciplinary Digital Publishing Institute |
| **VSA** | Vector-Symbolic Architectures |

## ACKNOWLEDGEMENTS

We would like to acknowledge the use of AI in refining the clarity and readability of this manuscript. The AI assistance was primarily used for tasks such as sentence restructuring, word choice suggestions, and identifying potentially unclear phrasing. We emphasize that the AI was used solely for language enhancement and did not contribute to the generation of research ideas, data analysis, or the interpretation of results. All conclusions drawn and insights presented in this manuscript are solely the product of the authors' own analysis and expertise.

### Funding

Davide Chicco has been financially supported by the Italian Ministero Italiano delle Imprese e del Made in Italy under the Digital Intervention in Psychiatric and Psychologist Services (DIPPS) (project code F/310240/01-04/X56) programme within the framework "Innovation Agreements" (Accordi per l'Innovazione) and by Ministero dell'Università e della Ricerca of Italy under the "Dipartimenti di Eccellenza 2023-2027" ReGAInS grant assigned to Dipartimento di Informatica Sistemistica e Comunicazione at Università di Milano-Bicocca. There was no additional external funding received for this study. The funders had no role in study design, data collection and analysis, decision to publish, or preparation of the manuscript.

### Grant Disclosures

The following grant information was disclosed by the authors:
Digital Intervention in Psychiatric and Psychologist Services (DIPPS): F/310240/01-04/X56.
Ministero dell'Università e della Ricerca of Italy: "Dipartimenti di Eccellenza 2023-2027".

## Competing Interests

Fabio Cumbo and Davide Chicco are Academic Editors for PeerJ Computer Science.

## Author Contributions

- Fabio Cumbo conceived and designed the experiments, performed the experiments, analyzed the data, performed the computation work, prepared figures and/or tables, authored or reviewed drafts of the article, and approved the final draft.
- Davide Chicco conceived and designed the experiments, performed the experiments, analyzed the data, performed the computation work, prepared figures and/or tables, authored or reviewed drafts of the article, and approved the final draft.

## Data Availability

This is a literature review.

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
