# Peer review of "Hyperdimensional computing in biomedical sciences: a brief review"

_PeerJ Computer Science, doi:10.7717/peerj-cs.2885_

## Round 0.1 · original submission · Major Revisions

Dear Authors,

You have appealed, and the Section Editors have agreed to allow the Appeal. Reviewers have concerns regarding the literature review and added value. We do encourage you to address the concerns and criticisms of the reviewers and resubmit your article once you have updated it accordingly. A more in-depth examination of the opportunities, pitfalls, strengths and weaknesses of the subject will also be facilitated.

Best wishes,

Reviewer 1 ·

Basic reporting

In this review, the authors scanned the literature for papers within biomedical sciences (bioinformatics, chemoinformatics etc) that use HDC/VSA. Of the forty papers they found, they performed a detailed analysis of the specific architecture, topic, journal, number of authors, hardware, programming language etc.

I think the authors have missed the following review in PLOS Computational Biology, which covers the same topic.

Stock M, Van Criekinge W, Boeckaerts D, Taelman S, Van Haeverbeke M, Dewulf P, et al. (2024) Hyperdimensional computing: A fast, robust, and interpretable paradigm for biological data. PLoS Comput Biol 20(9): e1012426. https://doi.org/10.1371/journal.pcbi.1012426

Personally, I find the manuscript to not have much added value. The authors do not give an introduction to HDC, which can act as a starting point. So, I don't think biomedical researchers are the key audience.

The collection of literature is analysed in depth, though I am missing a bit of a visionary perspective of the authors and what the gaps and potentials are. For example, much is done on CPU, but should research be done on FPGA? How does this subsection of HDC literature compare to the field as a whole? I believe a good review should not only list the existing work but also build a narrative that can be a basis to move the field forward or be an entry point for novices.

I think the remarks on reproducible science and peer review are relevant, though they may apply to computer science as a whole.

To summarize, this review would need to contain a general introduction to HDC and a discussion with opportunities, pitfalls, strengths and weaknesses to be qualitative.

Experimental design

The authors go in-depth into how they collected the papers for their literature survey. Here, they follow established protocols, which should be applauded.

My main, maybe more general, problem with these types of reviews is that they do not provide a lot of added value compared to online tools, which can generate the same report automatically (Elicit, Litmaps, etc.).

Validity of the findings

All the results seem to be correct interpretations.

Cite this review as

·

Basic reporting

This is an interesting survey that focuses on the biomedical side of hyperdimensional computing (HDC).

The number of references is 48; I believe there are more references in HDC literature in the bio-science field (even all other types -applications, theory- of literature to be used for the background section + biomedical & bioinformatics concepts). So, I believe that the authors should increase their total references (not only from the bio-science but others as well).

Also, I recommend that the authors look into recent works from both the venues and the open-source manuscript platforms, including 2024.

The motivation in the background should answer this question: "Why should a reader take this paper to read instead of/in addition to previous surveys, including the biomedical-related discussion?"

Experimental design

The reader expects a better background section. Some of the concepts are not technically included. For instance, a general picture of an HDC framework, including the modules inside for binding, bundling, shifting, permutation, etc. How the learning task is being done in the HDC? What are the similar or different learning concepts in HDC versus Neural Networks? Lines starting from 105: the authors should use some literature checks to include references plus more technical explanations.

The expression "...usually with 10-thousand dimensions" -> This is not always the case; with some new vector generation sources, such low discrepancy sequences (Sobol, Van der Corput-based quasi-randomness, instead of pseudo-randomness), this number reduces. I recommend that the authors do a quick literature review on this to extend this part of the manuscript. The ISO-accuracy concept allows showing how much-reduced vector size (D) can be in different generation-encoding scenarios when using other random sources, but for the same accuracy target. (E.g., 80% classification accuracy with D=1024 for baseline method, but D=968 with a better vector generation method for the same 80% accuracy. D is hypervector size.)

Figure 1 should update the year to 2024 as well. More figures and tables to compare the technical side of the works would be better.

The authors discuss binary-bipolar encoding; this is important but relatively less important than the encoding style, especially the generation of the vectors and their random sources.

Different learning concepts (e.g., single-pass learning), training strategies, and machine learning-related parameters during learning can be presented using biomedical HDC papers. Each of these efforts' learning schemes can be discussed in depth.

Validity of the findings

HDC's main target is to have lightweight hardware; the manuscript should also extend this side of HDC.

The manuscript is good at classifying the previous state-of-the-art papers with different considerations, such as venues, author names, etc. However, I am not sure how much it would be beneficial for the reader. This should be clarified in the manuscript. If this had been prepared for all HDC literature, then it may have brought a good discussion. The more in-depth technical side of the HDC & biomedical science should be discussed in the manuscript. What are the advantages of HDC in biomedical science, and how much gain-loss in accuracy and hardware efficiency do we have over all these works?

The authors can also discuss the limitations of these works. How can we connect different works to each other, considering their limitations?

Overall, the survey idea is good for this specific application, and I thank the authors for preparing this interesting work. Nevertheless, I recommend revising the points mentioned above, and then I believe the work will be stronger.

---

## Round 0.2 · Major Revisions

Dear Authors,

Thank you for submitting your article. Feedback from the reviewers is now available. It is not recommended that your article be published in its current format. However, we strongly recommend that you address the issues raised by the reviewers, especially those related to readability, experimental design and validity, and resubmit your paper after making the necessary changes.

Best wishes,

Reviewer 1 ·

Basic reporting

In general, I think the authors have made a good effort in strengthening their work. I think the additional context, explanation and interpretation of the paper has improved the paper a lot. However, there are still some points that could be improved.

1. In general, I think the introduction goes a little overboard, emphasizing the need for this survey. I think having a general introduction to HDC (from line 100 onwards) within biomedical sciences would be more appropriate. The end of the introduction would likely be a good place to state the aim and scope of the review and point to complementary works.
2. The operations from l160 are useful. It is probably best to call a) binding and b) aggregation/bundling as these are the more general terms (multiplication and addition do not make sense for binary vectors). The superposition principle could be explained a bit more clearly.
3. l 509 and 600: though the sentiment about preprints is likely just a caution, maybe a reference is in order to show that preprints really are less reliable than peer-reviewed work? I am not sure CS journals are less stringent than journals.
4. Especially l 443 contains a paragraph with quite heavy accusations. To me, this questions the scientific validity of dozens of papers without clear support.
5. l 540: maybe elaborate on the interpretability of HDC?
6. (discussion) Do the authors then also advocate standardized software, such as Scikit-learn to allow for a common way for HDC?

Experimental design

The authors seem to have taken care of performing a correct and comprehensive literature survey.

I think Figure 2 might be improved by sorting A based on prevalence. It would also be interesting to see the number of independent papers (not sharing a co-author) per topic. For example, is seizure detection mainly done by the same number of researchers? I would do this by subdividing the bar in a stacked plot, with different colors per group, or a second bar with a number of independent articles.

Validity of the findings

I think some general statements that question the rigor of papers not peer-reviewed might be somewhat unsupported. In my opinion, this should be supported with some references that show that conference articles are of a lower quality or be softened.

The section on "The Future of HDC" does not contain any references for the claims they make. Authors should at least cite a work that supports the claim, ie give an example.

Additional comments

- l 25, "Here we introduced" => "Here, we introduce"

Cite this review as

·

Basic reporting

The manuscript improved significantly. I thank the authors for the revisions and the response document.

No more comments for the basic reporting side of the manuscript.

Experimental design

The manuscript's discussions and background would benefit from a stronger connection to the broader literature. While citing biomedical papers is important as this is a biomedical HDC survey, many state-of-the-art works discuss the theory of HDC -e.g., "Key principles and advantages of Hyperdimensional Computing" section- in depth, with different approaches discussed in the beginning part of the survey. Incorporating these references would strengthen the discussion.

Additionally, while valuable focusing on publisher names, or researchers in HDC (Dr. Rosing, Dr. Imani, Dr. Rahimi, ...), who are already well-known and accessible also from the reference list, it would be more valuable to highlight key theoretical insights from these people. E.g., who is leaning toward which biomedical content; DNA, seizure detection, ... etc., over these number of papers -with one more column to the given list-

Validity of the findings

It would be helpful to clarify the definition of "randomness" across the works cited. What is the source of randomness, and how does it impact (1) accuracy and (2) hardware efficiency especially for the biomedical applications? Vector generation + encoding is the most important and expensive part of an HDC system; discussion on this is very important, especially considering the biomedical applications. The inclusion of hardware platforms is valuable. However, it would also be beneficial to discuss any hardware efficiency in the context of biomedical implementations, given that HDC is inherently a hardware-related computing paradigm.

Additional comments

Overall, it is an interesting survey in the HDC domain.

---

## Round 0.3 · accepted · Accept

Dear Authors,

Thank you for addressing the reviewers' comments. Your manuscript now seems sufficiently improved and ready for publication.

Best wishes,

Reviewer 1 ·

Basic reporting

check

Experimental design

check

Validity of the findings

check

Additional comments

check

Cite this review as

·

Basic reporting

I thank the authors for their revision; my concerns have been solved.

Experimental design

I thank the authors for their revision; my concerns have been solved.

Validity of the findings

I thank the authors for their revision; my concerns have been solved.